# Development of a Head Injury Criteria-Compliant Aircraft Seat by Design of Experiments

**Giuseppe Lamanna** [1], **Amalia Vanacore** [2], **Michele Guida** [2,*], **Francesco Caputo** [1], **Francesco Marulo** [2], **Bonaventura Vitolo** [3] and **Salvatore Cicatiello** [3]

[1] Department of Industrial and Information Engineering, University of Campania Luigi Vanvitelli, Aversa, 81100 Caserta, Italy

[2] Department of Industrial Engineering, University of Naples "Federico II", 80125 Napoli, Italy

[3] Geven, Zona Asi Boscofangone, 80035 Nola, Italy

* Correspondence: michele.guida@unina.it

**Abstract:** This paper deals with the redesign of an aircraft passenger seat, placed at the first seat row, which was not compliant with Federal Aviation Regulations FAR 25.562 "Emergency landing dynamic conditions" regulation (due to a high value for the Head Injury Criterion (HIC)) and related guidelines. Starting from an accurate analysis of some results obtained via an experimental seat sled test, a numerical procedure was developed in order to improve the passenger safety with respect to head injury. Specifically, the proposed numerical procedure, using the advantages of a Finite Element (FE) model and a Design of Experiment (DoE) approach for simulation modeling, was aimed at identifying a new design solution to avoid the impact between the passenger's head and the bulkhead. The redesign of the passenger seat was validated against an experimental test carried out at Geven S.p.A. Company by demonstrating, consequently, the compliance of the modified seat-belt system with the regulations.

**Keywords:** multibody systems; finite element analysis; aircraft seat; design of experiment; head injury

## 1. Introduction

Crashworthiness requirements drive the design of most automotive and aerospace structural components in order to improve their passive safety performance and, consequently, to protect occupants against injuries and death during a crash event. Design efforts are increasingly focussed on satisfying these requirements, even if passive safety criteria lead to a small increase in the structure's weight. Many research studies involving experimental tests and numerical studies are carried out to verify that the proposed design solutions respect crashworthiness requirements and to predict the injury level for occupants in case of an accident [1,2].

In the aeronautical sector, the sled is considered as an appropriate system to check the protection of occupants, and a crashworthy seat has to guarantee that the consequences of occupant injuries are not fatal [3]. Currently, aeronautical regulations consider the experimental test to be the only possible solution to certify an aircraft seat. For lighter aircraft (take-off weight of 5670 kg or less), two tests are required [4]: the first test is based on the loads transferred to the occupant during a forward deceleration; the second test considers the downward deceleration. For heavier aircraft (take-off weight above 5670 kg) [5], a test related to the deformation of the floor is required in addition to the previous two tests. Whatever the airplane category, it is necessary to evaluate the entity of the damage in the presence of a head impact. For this purpose, the experimental test for component certification can take advantage of the high level of technology introduced in the current Anthropometric Test Devices (ATDs), which allow the recording of the occupant's response (i.e., kinematical as well as

dynamical parameters) during the experimental simulation of a crash event [6]. However, it is evident that the fitting and the implementation of these tests are time-, space- and cost-demanding, and thus it is not affordable to experimentally test several configurations of a proposed design solution. For all these reasons, the development of a numerical model capable of reproducing such complex experimental tests, according to the Certification by Analysis paradigm, seems to be the best strategy for the development of a crashworthy design solution.

The main advantage provided by numerical methods consists in the possibility to identify an optimal virtual solution that can be validated via experimental tests, avoiding the "trial and error" approach typically adopted by companies and, consequently, reducing time and cost for development. Of course, the development of a reliable numerical model requires many efforts in dealing with the underlying assumptions and hypotheses as well as in the necessary availability of high computational power, complemented by high skills in modeling.

Among the several structural components that can significantly contribute to occupant safety, the seat plays a key role in reducing the loads transferred to passengers during the crash as well as in reducing the probability of the passenger's head impact against the bulkhead, in the case of the first seat row. In [6], the authors propose a multibody model of the aircraft seat structure for the simulation of a 16-g compliance with the Head Injury Criteria (HIC) requirement. This test involves an impact against a bulkhead developed and analyzed by using an ad hoc algorithm implemented in MATLAB® code, as a 2D system of rigid bodies interconnected by springs and joints. Lankarani, in [7], focuses on the design and development of bulkheads evaluating various honeycomb materials for HIC attenuation, rather than on high costs and schedule overruns due to the development and certification of aircraft seats.

The aim of this paper is to define a numerical procedure to improve the design of an aircraft passenger seat, considering passive safety as the main goal. For this purpose, different numerical methods can be used, including Finite Element (FE) and Multibody (MB) models.

MB numerical models are generally adopted to evaluate the kinematics of the dummy (representing the occupant) and its interactions with both the restraint systems, if available, and the seat cushion, with no possibility whatsoever to obtain any results about the structural behavior of the seat frame. These models exhibit the convenience of allowing quick modifications of the analyzed configuration and obtaining reliable results—within the recalled limits—with very short runtimes.

Numerical FE models are, instead, required to obtain information about the structural behavior of the seat. Even if the developing time of these models is considered acceptable for current design time scheduling, the same cannot be said for runtimes. In particular, runtimes are very long for applications of dynamical type and for the use of anthropomorphic dummies, whose models prove to be very complex and therefore require very accurate discretization and high mesh densities. Some solutions involving the coupling of both methods are proposed in literature [8,9].

The research activity presented in the present paper makes use of the FE and MB methods either by an internal explicit code (hybrid) [8] or independent code (coupling) [9], with a difference in terms of computational cost and correlation with experimental data. Specifically, the research starts from an established FE model presented by authors in previous papers [8]. This FE model allows the description of the kinematics of a passenger as well as the injuries that a passenger seated at the first row of an aircraft may suffer in a frontal impact against the bulkhead. A corresponding biomechanical head injury index was calculated and compared with the extreme one considered according to the current regulations. The investigation was also experimentally carried out by launching an aircraft seat, equipped with an ATD, against a bulkhead at the required speed and the corresponding acceleration/deceleration profile. The experimental data were used to calculate a damage index, as described below, useful for the evaluation of the level of injuries affecting different body parts directly involved in the impact or just subjected to high inertia loads. The good level of accuracy of the developed FE model was demonstrated by comparing numerical results to experimental data.

According to the experimental test presented in [8], the calculated HIC (Head Injury Criterion) parameter, representing the head injury, was higher than the limit provided by AC 25.862 and SAE 8049b because the head of the dummy contacted the rigid bulkhead. The analysis of both numerical results and experimental data suggested that this problem could be related to the stiffness characteristics of some seat components, which have been thus selected as design factors to be further investigated. Based on the validated FE model and the adoption of an efficient experimental design, new numerical experiments were carried out in order to find a new design solution for the seat frame to improve the passenger passive safety. All analyses were carried out using Ls-Dyna® software. In the following sections, the baseline experimental study, the FE model, the strategy for planning the numerical experiments and the obtained results are described and discussed.

## 2. Baseline Study and Dynamic Testing

A metallic seat frame fabricated in aluminum alloy was used in the dynamic sled test. To maximize the energy transferred to the head, no yaw was given to the seat in these tests. The experimental setup of the seat, shown in Figure 1, is representative of a typical airline economy class seat with the seat back fixed in the upright position.

The seat setback distance—defined as the horizontal distance between the seat reference point (i.e., the intersection point between the seat back and the seat pan) and the outer surface of the bulkhead—was fixed at 583 mm (23 inches).

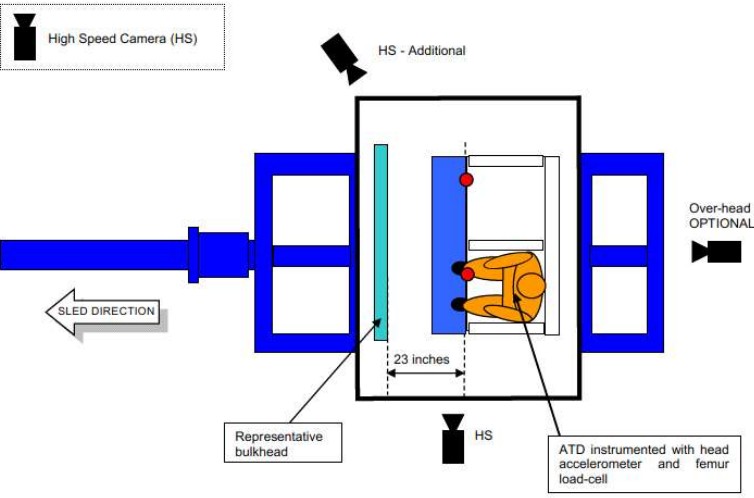

**Figure 1.** Experimental setup.

The seat cushions used during the test consisted of foam with the static deformation under the passenger weight set to a maximum penetration of 5 mm; a typical polyester seat belt was used. The sample full-scale sled test setup with the bulkhead is shown in Figure 2. A triaxial accelerometer was mounted at the center of gravity (c.g.) of the ATD head to determine the resultant head acceleration.

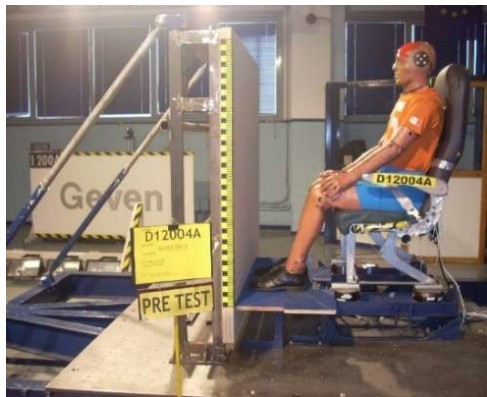

**Figure 2.** Full-scale sled test setup.

The experimental test sessions described in this section were conducted at the Impact Dynamics Laboratory of Geven S.p.A. with the aim of measuring the head accelerations of a Hybrid-II Anthropomorphic Test Dummy (ATD), as specified in Certification Specifications CS 25.562, [5], so as to support the development of the FE model and to assess the performances of the new seat design solution.

The experimental tests were carried out using a sled deceleration system pneumatically activated like a "sling". The seat with a dummy is fixed on the sled, and both together are launched at an assigned speed. The race is stopped by a mechanical brake (decelerator system) made with a lower carbon steel wire, which is properly assembled in order to have a g-peak compliant with the rules. The number, position, and length of each steel wire can affect the deceleration pulse shape. This HIC test can be performed in two different setup configurations:

- First row test, in which the seat is placed in front a rigid bulkhead at a fixed distance in order to simulate a typical first row installation inside the cabin (Figure 2). In this case, the main purpose is to verify the head contact, while the obtained HIC value can be considered only as a reference value since the installed bulkhead is not the real cabin installation.
- Row to row test, in which two seat rows are fixed on the sled at a proper distance. The aim of the test is measuring the HIC value during the head impact in order to evaluate the potential injury related to the design of the seat backrest including mounted equipment (monitor, rear table etc.).

As for the United States Code of Federal Regulations (CFR), a triangular deceleration pulse with a peak of 16 g and a rise time of 90 ms was targeted for the sled tests. The ideal pulse shape and the actual sled test deceleration pulse for the two baseline tests are shown in Figure 3. Proper seat belt installation required a test rig able to guarantee the correct position of aircraft/belt interface points with respect to the seat.

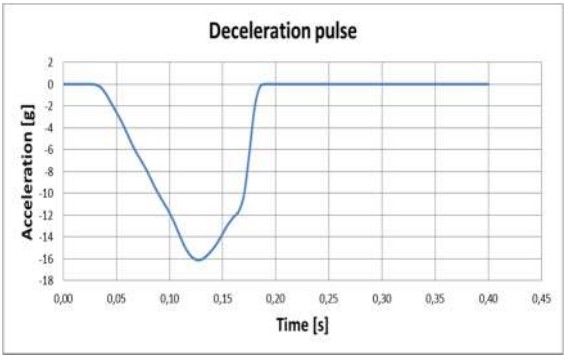

**Figure 3.** Deceleration pulse.

All devices were equipped with accelerometers and load cells to measure forces and acceleration affecting the most sensitive human body parts. Particular attention was paid to the acceleration of the head and to the loads transmitted to the lower limbs of the dummy in order to verify the accuracy of the FE model proposed in the previous papers [9,10] whose main characteristics are briefly described in the next section.

## 3. Numerical Models

The full seat FE model consisting of 105,226 elements and 151,219 nodes is shown in Figure 4. All structural seat components were modelled by considering aluminum alloy material, whereas foam material was considered for cushions, ref. [11].

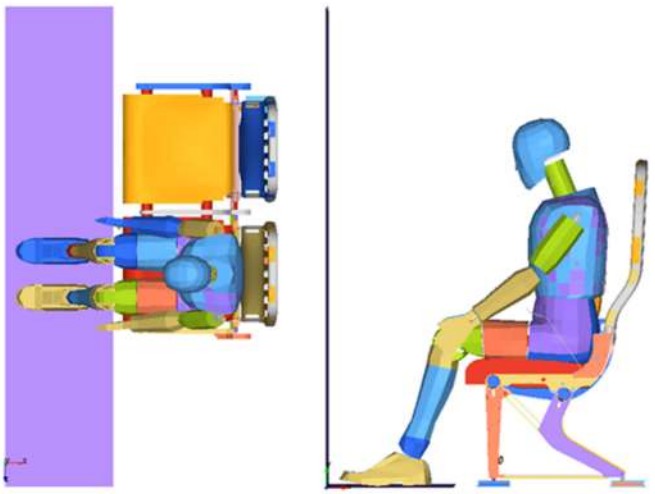

**Figure 4.** Finite Element (FE) model.

For each material, an elasto-plastic model was selected; the constitutive curves of each material are shown in Figure 5. The aluminum alloy adopted is an elasto-plastic material with kinematic hardening (model n. 24 Ls-dyna's material library), and failure is defined based on the plastic strain. The foam material considered is a rubber-like foam of polyurethane. It is a simple one-parameter model with a fixed Poisson's ratio of 0.25.

The deceleration pulse was applied to the nodes of both the bulkhead and the seat fixed to the slide. Gravity and initial velocity were applied to all parts of the model.

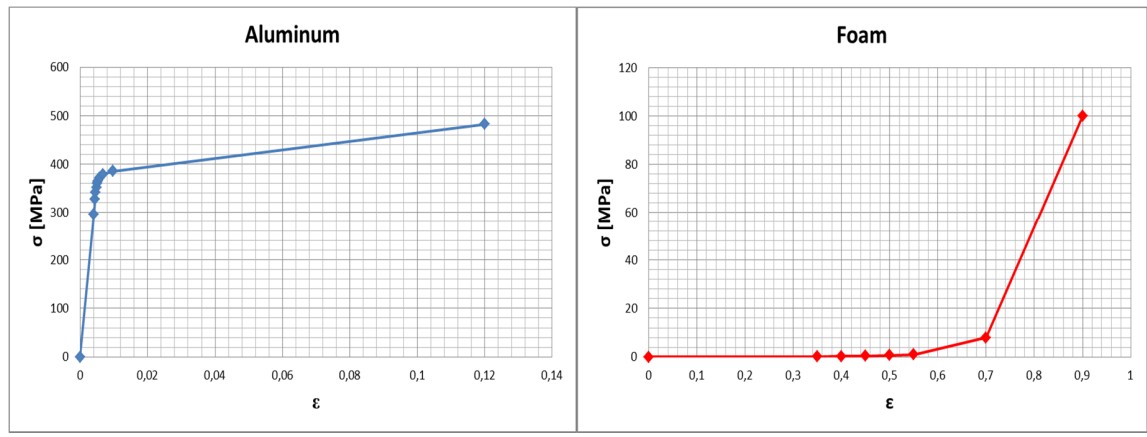

**Figure 5.** Sigma-epsilon curves.

The head acceleration curve, filtered according to [5], obtained from the FE model was compared against the head acceleration curve obtained from the sled test; both curves are shown in Figure 6.

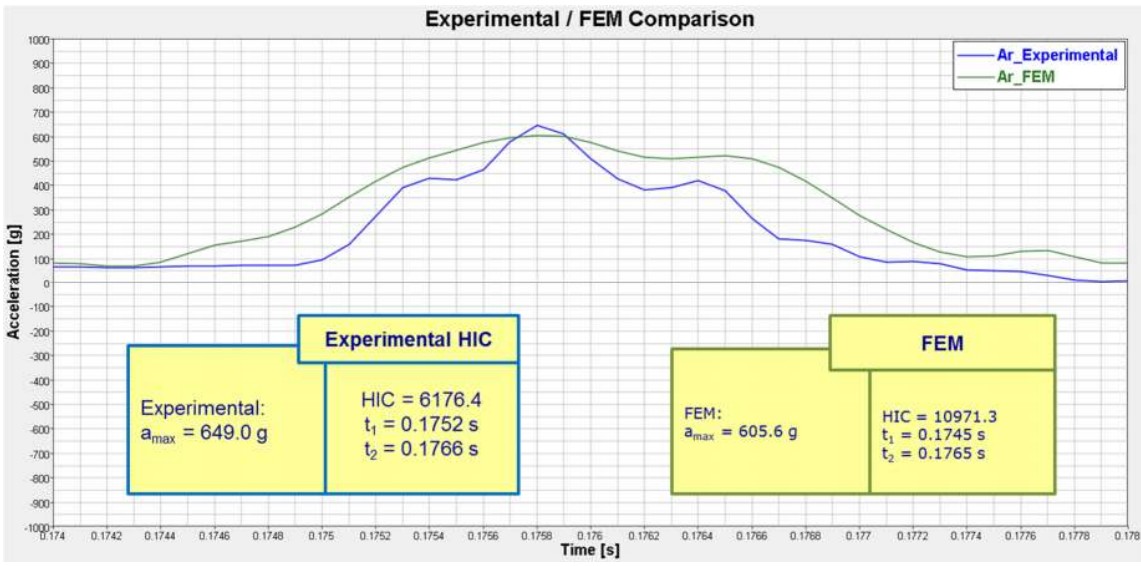

**Figure 6.** Experimental-numerical Head Injury Criterion (HIC) comparison.

A good agreement was achieved in terms of both acceleration peak and curve trend. Based on the numerical and experimental acceleration curves, the respective HIC values were calculated as follows:

$$HIC = \max\left\{ (t_2 - t_1) \left[ \frac{1}{(t_2 - t_1)} \int_{t_1}^{t_2} a(t)dt \right]^{2.5} \right\} \tag{1}$$

where $a(t)$ is the resultant head acceleration measured in $g$, and $t_1$ and $t_2$ are the extremes of the integration interval containing the head acceleration peak, measured in seconds.

Despite the good agreement between the acceleration peaks of the numerical and experimental head acceleration curves, the calculated HIC values were different. The difference in the HIC values can be explained taking into account that HIC is calculated choosing the extremes of the integration interval considering the whole acceleration head curve. Actually, as shown by Figure 7, the numerical and experimental curve trends are quite similar, but the corresponding areas are different.

The validated FE model was used to conduct a second round of numerical experiments to investigate the stiffness characteristics of some seat components to obtain useful information for re-designing the seat to avoid head–bulkhead impact. Only a single modification was introduced in the FE model: the belt was replaced with a Y-belt (Figure 7).

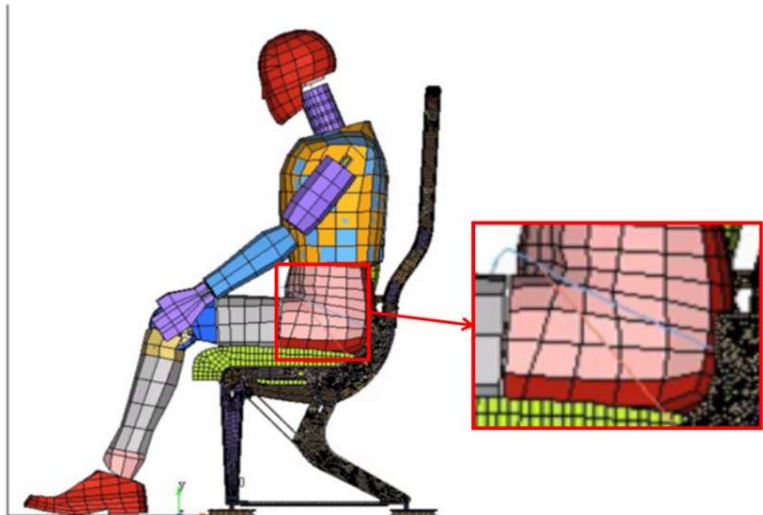

**Figure 7.** New FE model with modified belt.

## 4. Design of Numerical Experiments

Numerical experiments were planned according to a Plackett–Burman (PB) design [12] that is a two-level non-regular orthogonal design used as a highly resource-efficient strategy for planning experiments under the assumption of negligible interactions between design factors.

In recent years, the PB design has been proposed as an efficient strategy for factor screening; this was also realized in the specialized literature on crashworthiness. M. Hatami [13] used the Design of Experiment (DoE) method to perform optimization, setting the partial shape of a variable turbocharger using several parameters, and verified the effect. Zhang et al. in [14] approached the optimization design of a motorcycle engine.

Tarlochan and Faridz [15] considered the potential factors that could contribute to the frontal crash performance with the use of a 12-run PB design; Pradeep Kanna et al. [16] discussed vehicle structure behavior in a roof crush considering 16 factors in a 20-run PB design; for crashworthiness optimization of a vehicle body, Hou et al. [17] used a 20-run PB to study 19 factors on three responses; for vehicle side impact, Hou et al. [18] used a 20-run PB to study 19 factors and then 15 factors; Wang et al. [19] implemented a crashworthiness analysis of a vehicle door based on a PB design. Lin and Draper [20–22] showed that the projection of the PB design into a lower dimensional space corresponding to $k$ important factors leads to the identification of a helpful additional run.

For the PB design adopted in our study, addressing the head–bulkhead impact in relation to the seat frame and the belt, nine factors were taken into account: X and Y coordinates of the Anchor point ($AP_X$ and $AP_Y$, respectively); the thicknesses of the Rear beam ($D_1$), Rear leg frame ($D_2$), Rear leg web ($D_3$), Shock absorber ($D_4$) and Reinforcing beam ($D_5$); the width of the Rear leg frame ($D_6$) and the length of the Reinforcing beam ($D_7$). For cases of eight to eleven factors, the number of experiments, for a PB plan, is 12. However, in order to arrange the plan, all the eleven factors are necessary. The last two columns are two dummy factors ($dumm_1$ and $dumm_2$, respectively).

Two different levels are representative of the factors for the given experiment; one is relative to its high level and the other one denotes the factor at its low level. The design factors and their ranges are reported in Table 1 and shown in Figure 8. The response variable is a binary performance indicator representing the CONTACT/NO-CONTACT between the passenger head and the bulkhead.

**Table 1.** Design factors and ranges.

| Code | Design Factor | Range (mm) |
|---|---|---|
| $AP_X$ | Anchor Point (coord_X) | 26 ÷ 45 |
| $AP_Y$ | Anchor Point (coord_Y) | 44 ÷ 88 |
| $D_1$ | Rear beam (thick.) | 2 ÷ 3 |
| $D_2$ | Rear leg frame (thick.) | 3 ÷ 4.5 |
| $D_3$ | Rear leg web (thick.) | 1.8 ÷ 2.7 |
| $D_4$ | Shock abs (thick.) | 2 ÷ 3.2 |
| $D_5$ | Rear leg frame (width) | 18 ÷ 20 |
| $D_6$ | Reinforcing beam (thick.) | 1 ÷ 1.5 |
| $D_7$ | Reinforcing beam (length) | 150 ÷ 250 |
| dumm1 | DUMMY | 0 ÷ 1 |
| dumm2 | DUMMY | 0 ÷ 1 |

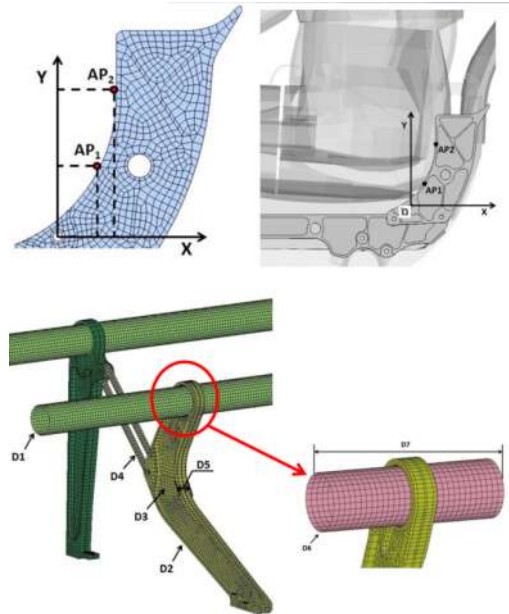

**Figure 8.** Design factors.

The results of the second round of numerical experiments, reported in Figure 9, show that the head–bulkhead impact is only avoided at run 2 with a distance of 2 mm between the top point of the passenger head and the bulkhead.

| Run | Ap_x | Ap_y | d1 | d2 | d3 | d4 | d5 | d6 | d7 | dum1 | dum2 | U |
|---|---|---|---|---|---|---|---|---|---|---|---|---|
| 1 | - | + | - | - | - | + | + | + | - | + | - | X |
| 2 | - | - | + | - | - | - | + | + | + | - | + | 2 mm |
| 3 | + | - | - | + | - | - | - | + | + | + | - | X |
| 4 | - | + | - | - | + | - | - | - | + | + | + | X |
| 5 | + | - | + | - | - | + | - | - | - | + | + | X |
| 6 | + | + | - | + | - | - | + | - | - | - | + | X |
| 7 | + | + | + | - | + | - | - | + | - | - | - | X |
| 8 | - | + | + | + | - | + | - | - | + | - | - | X |
| 9 | - | - | + | + | + | - | + | - | - | + | - | X |
| 10 | - | - | - | + | + | + | - | + | - | - | + | X |
| 11 | + | - | - | - | + | + | + | - | + | - | - | X |
| 12 | - | - | - | - | - | - | - | - | - | - | - | X |

| FACTOR | VALUE |
|---|---|
| Ap_x | 26 |
| Ap_y | 44 |
| d1 | 3 |
| d2 | 3 |
| d3 | 1.8 |
| d4 | 2 |
| d5 | 20 |
| d6 | 1.5 |
| d7 | 250 |
| dum1 | 0 |
| dum2 | 1 |

**Figure 9.** Results of the second round of numerical experiments.

Figure 9 shows the head displacements in two selected frames for both the preliminary and modified FE models, respectively.

According to Figure 10, the red displayed head trajectory is representative of the head–bulkhead impact in the preliminary FE model, whilst the green one indicates the lack of contact in the new model. It must be noticed that the developed FE model is very time-demanding, requiring about 20 h for each run. This suggests the development of a new model, able to take advantages from both the FE and the Multibody (MB) (less time-consuming) approaches. In a previous paper, [8], the authors demonstrated that computational costs are reduced by up to 2 h using a hybrid FE-MB model, which simultaneously implements MB models, for system components whose deformations do not influence the dynamic system responses and for which only kinematic aspects must be investigated (e.g., anthropomorphic dummies), and FE models for the other system components.

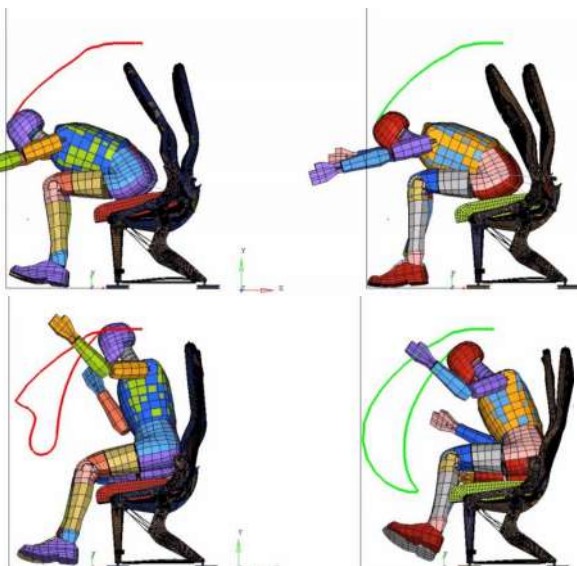

**Figure 10.** Head displacements for preliminary (on the left) and new (on the right) FE models.

## 5. Numerical-Experimental Comparison

An experimental test was carried out to verify the competence of the new restraint system as well as the proposed seat modifications. Figure 11 shows the new sled test configuration with details of the new Y-belt.

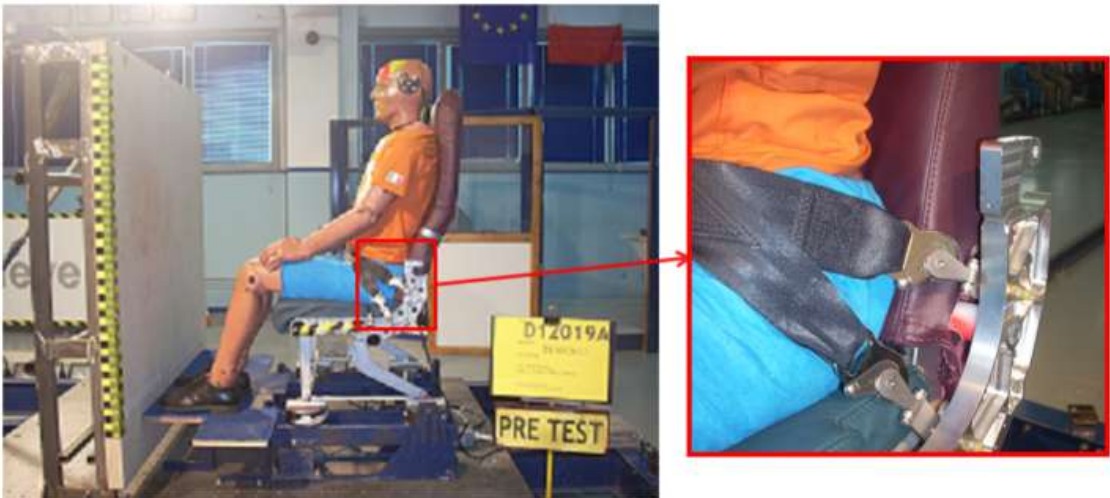

**Figure 11.** HIC front row experimental test.

The resultant deceleration profile applied to the sled is the same as the one used for the baseline experimental test described in Section 2 and shown in Figure 2. All baseline experimental conditions remained unchanged.

The result of the new experimental test was that the passenger head did not impact against the bulkhead, as predicted by the numerical simulation. Specifically, the distance between the bulkhead and the top point of the passenger head (at the maximum value of the head path due to a typical 16-g forward inertial loading condition) was experimentally evaluated to be 555 mm (21.8 inches). By avoiding the impact, it was possible to respect the HIC limit and to provide a passenger seat design configuration compliant with American FAR and European CS25.562 "Emergency landing dynamic conditions" regulation and related guidelines.

## 6. Conclusions

This paper deals with the re-design of an aircraft passenger seat in order to make it certifiable according to CS 25.562 "Emergency landing dynamic conditions" regulation and related guidelines. The results of previous experimental investigations and an established FE model for the simulation of a seat sled test demonstrated that the passenger seat did not comply with the regulations (providing an excessively high HIC value). Starting from such results, a new numerical procedure is presented in this paper to improve the passenger safety. Specifically, a highly resource-efficient strategy for planning numerical experiments is adopted for gathering useful information to redesign both the seat frame primary structure and the restraint system (belt) in order to avoid the impact between the passenger's head and the bulkhead. Working numerically on the belt anchor points, thicknesses of the rear beam, rear leg frame, rear leg web, shock absorber and reinforcing beam, width of the rear leg frame and length of the reinforcing beam, it was possible to propose a redesigned passenger-seat system able to avoid the impact under a deceleration of 16 g. The numerical solution was then validated experimentally at Geven S.p.A. laboratory.

Further investigation will be carried out in order to apply a DoE approach, based on a new numerical model combining both the MB and FE methods, in order to further reduce the computational costs. In this way, it will be possible to investigate more quickly the response of the passenger-restraint system.

**Author Contributions:** Conceptualization, M.G.; Methodology, F.C.; G.L.; Validation, G.L. and M.G. Investigation, M.G.; Resources, B.V. and S.C.; Data Curation, A.V. and F.M.; Writing-Original Draft Preparation, G.L. and M.G.; Writing-Review & Editing, G.L. and M.G.; Supervision, F.M., F.C., A.V.

**Funding:** This research received no external funding.

**Conflicts of Interest:** The authors declare no conflict of interest.

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
