# Peer review of "Development of a Head Injury Criteria-Compliant Aircraft Seat by Design of Experiments"

_aerospace, doi:10.3390/aerospace6090095_

Round 1
Reviewer 1 Report
Major:You write: “HIC is calculated over the area under the whole acceleration head curve.” This is wrong! Please correct this (the HIC is computed over a moving interval of either 36 ms or 15 ms, see the literature. If you used what you wrote your computations are wrong and you have to repeat them all. Which limit did you use for the HIC value?
The discussion of your results is not sufficiently deep. It is too weak for a journal paper just to state that you have done a DoE and found one good solution. The results should be of interest for the reader; hence, either you propose an interesting method or you give interesting structural insights / interesting applications. In your case, the method is standard, hence work on the application. I recommend that you extend the DoE by a sensitivity study; which parameters are influential and why? Can you reduce the number of parameters? You may even try a surrogate model.
Minor:
You write: “evaluate the entity of the damage in the presence of a head impact” – this is not precise enough; the HIC is based on the resultant acceleration measured in the CoG of the dummy head; there is no “damage”. Re-write this with more correct details.
You write: “A corresponding biomechanical head injury index was calculated and compared with the extreme one considered in the current regulations” – this is not clear enough; please give details and equations. Add as well more details for the statement below in the text (“some damage indexes”).
You write (page 3): “are launched, at a properly speed” – give the details; the approach should be repeatable; here, give the speed. In addition, you write “HIC test may be performed in two different set up configuration” – why “may”; write what was done.
The “elastic-plastic model” has to be described more in detail (for both, the aluminum and the foam). Which material model is chosen for aluminum, a von Mises model with strain-rate dependence? Which model is used for the foam? You only give a one-dimensional stress-strain curve for a three-dimensional model. The reader should be able to repeat the study.
“head acceleration curve”: explain where exactly this is measured. Did you use the CoG of the dummy head and did you compute the resultant of the acceleration vector? Did you use a filter for the numerical and/or for the experimental curve? The curves are very smooth (normally you get something with high oscillations).
For the HIC: clarify if this is the HIC measured over 36 ms or not. The HIC you have is too high. In automotive the thresholds are 1000 for the HIC-36 ms and 700 for the HIC-15ms. You have more than 6000. Again, I believe the way you computed the HIC is wrong.
Also concerning the max acceleration: your values are extremely high; in automotive they allow 80 g over 3ms and not 605 or 649 g. Comment on this. I recommend not to give the maximal value but the maximal value which lasted at least 3 ms (so-called a3ms value).
Give reasons why you chose a Plackett-Burman (PB) design. Make it clearer why you say that the experimental results match the simulations. Which method did you use to compare?
There is a high number of smaller corrections needed, see my comments in the pdf.
General:
In the total text, do not use long sentences; provide only one message in each sentence. Hence, please edit the text accordingly. Write in present tense; it is not a report of what was done but a suggestion how to do similar things. The reference list should be re-written (see comments in the pdf).

Author Response
Dear Reviewers,
thank you for your comments.
Question 1:
“HIC is calculated over the area under the whole acceleration head curve.” This is wrong! Please correct this (the HIC is computed over a moving interval of either 36 ms or 15 ms, see the literature. If you used what you wrote your computations are wrong and you have to repeat them all. Which limit did you use for the HIC value?
Answer:
Your comment is right as far as the automotive field is concerned. In the aerospace field the calculation of the HIC, for a head impact, is not based on a fixed time interval. The value is searched along the whole acceleration curve by varying the time interval searching of the maximum value. We can simplify the HIC formula introducing a family of curves for different values of d = t2 − t1. We then vary the value of d. The value of the highest peak of the family of curves obtained gives us the HIC. It is very common for aerospace application.
Question 2:
The discussion of your results is not sufficiently deep. It is too weak for a journal paper just to state that you have done a DoE and found one good solution. The results should be of interest for the reader; hence, either you propose an interesting method or you give interesting structural insights / interesting applications. In your case, the method is standard, hence work on the application. I recommend that you extend the DoE by a sensitivity study; which parameters are influential and why? Can you reduce the number of parameters? You may even try a surrogate model.
Answer:
Your comment is also very relevant here.
The choice of the design variables followed a long path of mechanical design carried out with the company Geven during two international research projects. The choice of the nine factors is the result of such long research work. The objective of the work is to define a highly efficient strategy in terms of resources for the redesign of an aircraft passenger seat, an objective that, with all the limitations highlighted by you, appears to be achieved.
Questions 3, 4, and 5:
Corrected
Question 6:
The “elastic-plastic model” has to be described more in detail (for both, the aluminum and the foam). Which material model is chosen for aluminum, a von Mises model with strain-rate dependence? Which model is used for the foam? You only give a one-dimensional stress-strain curve for a three-dimensional model. The reader should be able to repeat the study.
Answer: Ok, the details are reported in the line 155-158
Question 7:
“head acceleration curve”: explain where exactly this is measured. Did you use the CoG of the dummy head and did you compute the resultant of the acceleration vector? Did you use a filter for the numerical and/or for the experimental curve? The curves are very smooth (normally you get something with high oscillations).
Answer:
HIC calculation is compliant the ref. n. 5 as specified in the paper (line 119).
Question 8:
For the HIC: clarify if this is the HIC measured over 36 ms or not. The HIC you have is too high. In automotive the thresholds are 1000 for the HIC-36 ms and 700 for the HIC-15ms. You have more than 6000. Again, I believe the way you computed the HIC is wrong.
Answer:
See answer to question 1.
Question 9:
Also concerning the max acceleration: your values are extremely high; in automotive they allow 80 g over 3ms and not 605 or 649 g. Comment on this. I recommend not to give the maximal value but the maximal value which lasted at least 3 ms (so-called a3ms value).
Answer:
Your comment is also very relevant here.
As stated at the beginning of the paper we are facing an experimental test not in target. The main reason for the failure of such experimental test is precisely the too high value of the acceleration. However, our work consists in finding a highly efficient strategy in terms of resources for the redesign of an aircraft passenger seat, the numerical-experimental correlation is referred to the available experimental test.
Question 10:
Corrected
Question 11:
Corrected
Reviewer 2 Report
The work aims at identifying a novel design solution for an aircraft passenger seat, able to avoid the impact between the passenger’s head and the bulkhead. The problem is tackled numerically by means of the finite element (FE) and multibody (MB) models, with a reduced computational cost. The numerical solutions are verified against appropriate experimental tests. The work is very interesting and it can represent a key reference for the aerospace design purposes. The work is well written and organized, and it is suitable for publication. The authors are only invited to check for the systematic error related to the unrecognized reference sources. After making this minor revision, the work can be accepted for publication.
Author Response
Thanks a lot, in this new issue the authors completed your suggestions.

Round 2
Reviewer 1 Report
Unfortunately, not all my comments have been considered in the revision. Hence, I repeat them partially below (e.g. concerning material model and acceleration filter). Again, the references are very incorrect.
Major aspect:
1) The material models are still not sufficiently described. The reader must (!) be enabled to repeat your study. In lines 154-158, your write:
“For each material, an elastic-plastic model has been selected; …”
Which model, a von Mises model? With or without hardening? Associated or non-associated? Which type of hardening (isotropic/kinematic, linear/nonlinear)? You may give the LS-Dyna name of the model. Please give also the parameters like yield strength and hardening parameters. Add also the elastic parameters (Young’s modulus and Poisson ratio for the aluminum).
“… the constitutive curves of each material are shown in Figure 5.”
How did you obtain these curves? Are they from experiments or from literature?
“The aluminum, which is an elasto-plastic material with an arbitrary stress as a function of strain curve and arbitrary strain rate dependency is defined …”
You can skip this “with an arbitrary stress as a function of strain curve” because this is evident. I believe you did not include strain rate dependency because you only show a single curve, which is independent on the time derivative of the total strain. Hence, “arbitrary strain rate dependency” is wrong.
“… as well as failure based on a plastic strain….”
Which failure model did you use (again you may give the LS-Dyna name)? Which parameters did you use for the failure?
“… The foam material is considered a rubber like foams of polyurethane. It is a simple one-parameter model with a fixed Poisson’s ratio of .25…”
For me, a one-parameter model does not exist. Please give details.
Please realize also the following minor corrections:
1) I would put the references in brackets, i.e. [1], [2], etc.
2) Lines 31/32: Replace “as an appropriate system of safety to check the protection the occupants” by “as an appropriate system to check the protection of occupants”
3) Line 36: Replace “down deceleration” by “downward deceleration”
4) Line 92: Use “Head Injury Criterion”; criteria is plural.
5) Line 96: Delete one of the two “which have been”
6) Line 105: something is missing
7) Line 108: Replace “Seat setback distance” by “The seat setback distance”
8) Line 123: Replace “at a assigned speed” by “at an assigned speed”
9) Line 125: Replace “HIC test” by “This HIC test”
10) Line 164: Mention if a numerical filter was used (give the type of filter) to obtain the acceleration curve because the peak acceleration depends strongly on this. Your curves are very smooth, which is unusual.
11) Line 167: You write “acceleration curve obtained from the sled tests” – how many tests were performed? Did you average the results?
12) Line 167: Your write “good agreement …” - Make it clearer why you say that the experimental results match the simulations. Which method did you use to compare?
13) Line 177: Replace “experimental curves trends” by “experimental curve trends”
14) Caption of Figure 7: Replace “New FE model” by “New FE model with modified belt”
15) Line 187: Replace “also realized” by “was also realized”
16) Line 191: Use “behavior” and not “behaviour" (you chose American English and not British)
17) Line 200: Replace “number of run” by “number of experiments”
18) Line 209: Replace “does not occur only” by “is only avoided”
19) Figure 11: Just a question: is submarining also a problem in aerospace and is the y-type belt improving the situation concerning submarining risk? You may a comment in the text.
20) Lines 249/250: Replace “Starting from the results …” by “The results …”
21) Line 251: Replace “… sled test that demonstrated by numerical simulation that the …” by “…sled test demonstrated that the”
22) Line 259: Replace “a deceleration of 16G” by “a deceleration of 16 g”
23) Line 260: Replace “was then verified experimentally” by “was then validated experimentally” (verification has a different meaning, see the discussion of verification & validation V&V).
24) Change the references as follows:
[1], [2]: add date of access
[3]: Use the reference as given online: “Lankarani H.M. (1997) Current Issues Regarding Aircraft Crash Injury Protection. In: Ambrósio J.A.C., Pereira M.F.O.S., da Silva F.P. (eds) Crashworthiness of Transportation Systems: Structural Impact and Occupant Protection. NATO ASI Series (Series E: Applied Sciences), vol 332. Springer, Dordrecht”
[6]: Give the city/country as location of the conference
[8], [9], [10]: Give volume, series number, and pages. (for [10] the year is missing!)
[14], [15]: Where are these publications published?
[15]: Give all author names
Author Response
Unfortunately, not all my comments have been considered in the revision. Hence, I repeat them partially below (e.g. concerning material model and acceleration filter). Again, the references are very incorrect.
Major aspect:
1) The material models are still not sufficiently described. The reader must (!) be enabled to repeat your study. In lines 154-158, your write:
“For each material, an elastic-plastic model has been selected; …”
Which model, a von Mises model? With or without hardening? Associated or non-associated? Which type of hardening (isotropic/kinematic, linear/nonlinear)? You may give the LS-Dyna name of the model. Please give also the parameters like yield strength and hardening parameters. Add also the elastic parameters (Young’s modulus and Poisson ratio for the aluminum).
“… the constitutive curves of each material are shown in Figure 5.”
How did you obtain these curves? Are they from experiments or from literature?
“The aluminum, which is an elasto-plastic material with an arbitrary stress as a function of strain curve and arbitrary strain rate dependency is defined …”
You can skip this “with an arbitrary stress as a function of strain curve” because this is evident. I believe you did not include strain rate dependency because you only show a single curve, which is independent on the time derivative of the total strain. Hence, “arbitrary strain rate dependency” is wrong.
“… as well as failure based on a plastic strain….”
Which failure model did you use (again you may give the LS-Dyna name)? Which parameters did you use for the failure?
“… The foam material is considered a rubber like foams of polyurethane. It is a simple one-parameter model with a fixed Poisson’s ratio of .25…”
For me, a one-parameter model does not exist. Please give details.
Answer:
The seat is mainly made of a typical aerospace aluminum alloy, 2024-T3. It is a material well known in the literature whose mechanical characterization is not reported in this work. We have carried out several numerical simulations using the strain rate mechanical properties, obtaining very similar results. For this reason, in the present work we have decided to adopt a simplified model that does not take into account the strain rate effects on the mechanical properties. Also with regard to the rupture criteria inserted in the numerical model, it was decided to adopt a simplified model, inserting the elongation at break as the only criterion.
This is a type of material with kinematic hardening. The material model is the number 24 (ls-dyna's material library).
All information has been included in the new version of the paper.
Done.
Please realize also the following minor corrections:
1) I would put the references in brackets, i.e. [1], [2], etc.
Done.
2) Lines 31/32: Replace “as an appropriate system of safety to check the protection the occupants” by “as an appropriate system to check the protection of occupants”
Done.
3) Line 36: Replace “down deceleration” by “downward deceleration”
Done.
4) Line 92: Use “Head Injury Criterion”; criteria is plural.
Done.
5) Line 96: Delete one of the two “which have been”
Done.
6) Line 105: something is missing
Done.
7) Line 108: Replace “Seat setback distance” by “The seat setback distance”
Done.
8) Line 123: Replace “at a assigned speed” by “at an assigned speed”
Done.
9) Line 125: Replace “HIC test” by “This HIC test”
Done.
10) Line 164: Mention if a numerical filter was used (give the type of filter) to obtain the acceleration curve because the peak acceleration depends strongly on this. Your curves are very smooth, which is unusual.
Done.
11) Line 167: You write “acceleration curve obtained from the sled tests” – how many tests were performed? Did you average the results?
Done! One single test is performed!
12) Line 167: Your write “good agreement …” - Make it clearer why you say that the experimental results match the simulations. Which method did you use to compare?
In this research field (aerospace crashworthiness), many scientific publications, referring at least to the last twenty years, deal with impact phenomena where you find numerical experimental correlations. Most of those who deal with this topic (once read the description of the model and the test) consider the curves in object in good agreement.
13) Line 177: Replace “experimental curves trends” by “experimental curve trends”
Done.
14) Caption of Figure 7: Replace “New FE model” by “New FE model with modified belt”
Done.
15) Line 187: Replace “also realized” by “was also realized”
Done.
16) Line 191: Use “behavior” and not “behaviour" (you chose American English and not British)
Done.
17) Line 200: Replace “number of run” by “number of experiments”
Done.
18) Line 209: Replace “does not occur only” by “is only avoided”
Done.
19) Figure 11: Just a question: is submarining also a problem in aerospace and is the y-type belt improving the situation concerning submarining risk? You may a comment in the text.
Submarining can be a serious problem also in aerospace. The topic is long and complex and probably it's necessary another publication to deal with it.
20) Lines 249/250: Replace “Starting from the results …” by “The results …”
Done.
21) Line 251: Replace “… sled test that demonstrated by numerical simulation that the …” by “…sled test demonstrated that the”
Done.
22) Line 259: Replace “a deceleration of 16G” by “a deceleration of 16 g”
Done.
23) Line 260: Replace “was then verified experimentally” by “was then validated experimentally” (verification has a different meaning, see the discussion of verification & validation V&V).
Done.
24) Change the references as follows:
[1], [2]: add date of access
[3]: Use the reference as given online: “Lankarani H.M. (1997) Current Issues Regarding Aircraft Crash Injury Protection. In: Ambrósio J.A.C., Pereira M.F.O.S., da Silva F.P. (eds) Crashworthiness of Transportation Systems: Structural Impact and Occupant Protection. NATO ASI Series (Series E: Applied Sciences), vol 332. Springer, Dordrecht”
[6]: Give the city/country as location of the conference
[8], [9], [10]: Give volume, series number, and pages. (for [10] the year is missing!)
[14], [15]: Where are these publications published?
[15]: Give all author names
Done.